# Neural criticality from effective latent variables

**Mia C Morrell[1], Ilya Nemenman[2], Audrey Sederberg[3]\*†**

[1]Department of Physics, New York University, New York, United States; [2]Department of Physics, Department of Biology, Initiative in Theory and Modeling of Living Systems, Emory University, Atlanta, United States; [3]Department of Neuroscience, University of Minnesota Medical School, Minneapolis, United States

**\*For correspondence:**
audrey.sederberg@gatech.edu

**Present address:** †School of Psychology and School of Physics, Georgia Institute of Technology, Atlanta, United States

**Abstract** Observations of power laws in neural activity data have raised the intriguing notion that brains may operate in a critical state. One example of this critical state is 'avalanche criticality', which has been observed in various systems, including cultured neurons, zebrafish, rodent cortex, and human EEG. More recently, power laws were also observed in neural populations in the mouse under an activity coarse-graining procedure, and they were explained as a consequence of the neural activity being coupled to multiple latent dynamical variables. An intriguing possibility is that avalanche criticality emerges due to a similar mechanism. Here, we determine the conditions under which latent dynamical variables give rise to avalanche criticality. We find that populations coupled to multiple latent variables produce critical behavior across a broader parameter range than those coupled to a single, quasi-static latent variable, but in both cases, avalanche criticality is observed without fine-tuning of model parameters. We identify two regimes of avalanches, both critical but differing in the amount of information carried about the latent variable. Our results suggest that avalanche criticality arises in neural systems in which activity is effectively modeled as a population driven by a few dynamical variables and these variables can be inferred from the population activity.

## eLife assessment

This paper provides a simple example of a neural-like system that displays criticality, but not for any deep reason; it's just because a population of neurons are driven (independently!) by a slowly varying latent variable, something that is common in the brain. Moreover, criticality does not imply optimal information transmission (one of its proposed functions). The work is likely to have an **important** impact on the study of criticality in neural systems and is **convincingly** supported by the experiments presented.

## Introduction

The neural criticality hypothesis – the idea that neural systems operate close to a phase transition, perhaps for optimal information processing – is both ambitious and banal. Measurements from biological systems are limited in the range of spatial and temporal scales that can be sampled, not only because of the limitations of recording techniques but also due to the fundamentally non-stationary behavior of most, if not all, biological systems. These limitations make proving that an observation indicates critical behavior difficult. At the same time, the idea that brain networks are critical echoes the anthropic principle: tuned another way, a network becomes quiescent or epileptic and in either state, seems unlikely to support perception, thought, or flexible behavior, yet these observations do not explain how such fine-tuning could be achieved. Further muddying the water, researchers have reported multiple kinds of criticality in neural networks, including through analysis of avalanches

(*Beggs and Plenz, 2003*; *Plenz et al., 2021*; *O'Byrne and Jerbi, 2022*; *Girardi-Schappo, 2021*) and of coarse-grained activity (*Meshulam et al., 2019*), as well as of correlations (*Dahmen et al., 2019*). How these flavors of critical behavior relate to each other or any functional network mechanism is unknown.

The phenomenon that we will refer to as 'avalanche criticality' appears remarkably widespread. It was first observed in cultured neurons (*Beggs and Plenz, 2003*) and later studied in zebrafish (*Ponce-Alvarez et al., 2018*), turtles (*Shew et al., 2015*), rodents (*Ma et al., 2019*), monkeys (*Petermann et al., 2009*), and even humans (*Poil et al., 2008*). The standard analysis, described later, requires extracting power-law exponents from fits to the distributions of avalanche size and of duration and assessing the relationship between exponents. There is debate over whether these observations reflect true power laws, but within the resolution achievable from experiments, neural avalanches exhibit power laws with exponent relationships predicted from theory developed in physical systems (*Perkovic et al., 1995*).

Avalanche criticality is not the only form of criticality observed in neural systems. Zipf's law, in which the frequency of a network state is inversely proportional to its rank, appears in systems as diverse as fly motion estimation and the salamander retina (*Mora and Bialek, 2011*; *Schwab et al., 2014*; *Aitchison et al., 2016*). More recently, *Meshulam et al., 2019* reported various statistics of population activity in the mouse hippocampus, including the eigenvalue spectrum of the covariance matrix and the activity variance. These were found to scale as populations were 'coarse-grained' through a procedure in which neural activities were iteratively combined based on similarity. Similar observations have been reported in spontaneous activity recorded across a wide range of brain areas in the mouse (*Morales et al., 2023*). Simple neural network models of such data explain neither Zipf's law nor coarse-grained criticality (*Meshulam et al., 2019*).

Even though these three forms of criticality are observed through different analyses, they may originate from similar mechanisms. Numerous studies have reported relatively low-dimensional structure in the activity of large populations of neurons (*Mazor and Laurent, 2005*; *Ahrens et al., 2012*; *Mante et al., 2013*; *Pandarinath et al., 2018*; *Stringer et al., 2019*; *Nieh et al., 2021*), which can be modeled by a population of neurons that are broadly and heterogeneously coupled to multiple latent (i.e. unobserved) dynamical variables. Using such a model, we previously reproduced scaling under coarse-graining analysis within experimental uncertainty (*Morrell et al., 2021*). Zipf's law has been explained by a similar mechanism (*Schwab et al., 2014*; *Aitchison et al., 2016*; *Humplik and Tkačik, 2017*). A single quasi-static latent variable has been shown to produce avalanche power laws, but not the relationships expected between the critical exponents (*Priesemann and Shriki, 2018*), while a model including a global modulation of activity can generate avalanche criticality (*Mariani et al., 2021*), but has not demonstrated coarse-grained criticality (*Morrell et al., 2021*). It is not known under what conditions the more general latent dynamical variable model generates avalanche criticality.

Here, we examine avalanche criticality in the latent dynamical variable model of neural population activity. We find that avalanche criticality is observed over a wide range of parameters, some of which may be optimal for information representation. These results demonstrate how criticality in neural recordings can arise from latent dynamics in neural activity, without need for fine-tuning of network parameters.

## Results
### Critical exponents values and crackling noise

We begin by defining the metrics used to quantify avalanche statistics and briefly summarize experimental observations, which have been reviewed in detail elsewhere (*Plenz et al., 2021*; *O'Byrne and Jerbi, 2022*; *Girardi-Schappo, 2021*). Activity is recorded across a set of neurons and binned in time. Avalanches are then defined as contiguous time bins, in which at least one neuron in the population is active. The duration of an avalanche is the number of contiguous time bins and the size is the summed activity during the avalanche. The distributions of avalanche size and duration are fit to power laws ($P(S) \sim S^{-\tau}$ for size $S$, and $P(D) \sim D^{-\alpha}$ for duration $D$) using standard methods (*Clauset et al., 2009*).

Power laws can be indicative of criticality, but they can also result from non-critical mechanisms (*Touboul and Destexhe, 2017*; *Priesemann and Shriki, 2018*). A more stringent test of criticality

is the 'crackling' relationship (*Perkovic et al., 1995*; *Touboul and Destexhe, 2017*), which involves fitting a third power-law relationship, $\bar{S}(D) \sim D^{\gamma_{fit}}$, and comparing $\gamma_{fit}$ to the predicted exponent $\gamma_{pred}$, derived from the size and duration exponents, $\tau$ and $\alpha$:

$$\gamma_{fit} \stackrel{?}{=} \gamma_{pred} \equiv \frac{\alpha - 1}{\tau - 1}. \tag{1}$$

Previous work demonstrating approximate power laws in size and duration distributions through the mechanism of a slowly changing latent variable did not generate crackling (*Touboul and Destexhe, 2017*; *Priesemann and Shriki, 2018*).

Measuring power-laws in empirical data is challenging: it generally requires setting a lower cut-off in the size and duration, and the power-law behavior only has limited range due to the finite size and duration of the recording itself. Nonetheless, there is some consensus (*Shew et al., 2015*; *Fontenele*

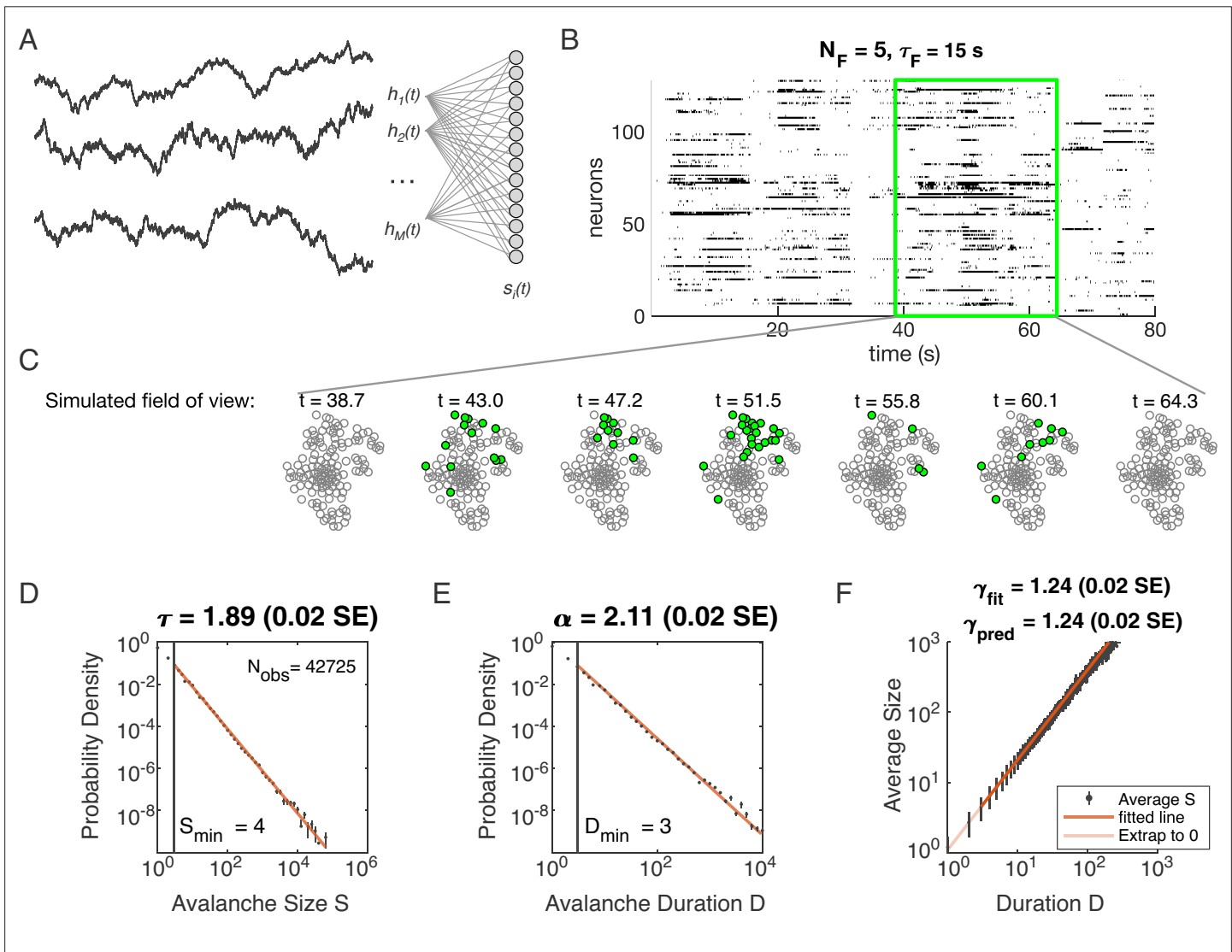

**Figure 1.** Latent dynamical variable model produces avalanche criticality. Simulated network is $N = 1024$ neurons. Other parameters in *Table 1*. (**A**) Model structure. Latent dynamical variables $h_\mu(t)$ are broadly coupled to neurons $s_i(t)$ in the recorded population. (**B**) Raster plot of a sample of activity binned at 3 ms resolution across 128 neurons with five latent variables, each with correlation timescale $\tau_F = 15$ s. (**C**) Projection of activity into a simulated field of view for illustration. (**D-F**) Avalanche analysis in a network (parameters $N_F = 5$, $\tau_F = 10^4$, $\eta = 4$ and $\epsilon = 12$), showing size distribution (**D**), duration distribution (**E**), and size with duration scaling (**F**). Lower cutoffs used in fitting are shown with vertical lines and their values are indicated in the figures. There are $N_{obs} = 42725$ avalanches of size $S \geq S_{min}$ in this simulated dataset. Estimated values of the critical exponents are shown in the titles of the panels.

**Table 1.** Simulation parameters for *Figure 1*.

| Parameter | Description | Value |
|---|---|---|
| $\epsilon$ | bias towards silence | $\epsilon = 12$ |
| $\eta$ | variance multiplier | $\eta = 4.0$ |
| $N_F$ | number of latent fields | $N_F = 5$ |
| $\tau_F$ | latent field time constant | $\tau = 10^4$ |
| $N$ | number of cells | $N = 1024$ |

*et al., 2019*; *Ma et al., 2019*) that even if $\tau$ and $\alpha$ vary over a wide range (1.5 to about 3) across recordings, the values of $\gamma_{\text{fit}}$ and $\gamma_{\text{pred}}$ stay in a relatively narrow range, from about 1.1 to 1.3.

## Avalanche scaling in a latent dynamical variable model

We study a model of a population of neurons that are not coupled to each other directly but are driven by a small number of latent dynamical variables – that is, slowly changing inputs that are not themselves measured (*Figure 1A*). We are agnostic as to the origin of these inputs: they may be externally driven from other brain areas, or they may arise from large fluctuations in local recurrent dynamics. The model was chosen for its simplicity, and because we have previously shown that this model with at least about five latent variables can produce power laws under the coarse-graining analysis (*Morrell et al., 2021*). In this paper, we examine avalanche criticality in the same model.

Specifically, we model the neurons as binary units ($s_i$) that are randomly ($J_{i\mu} \sim N(0, 1)$) coupled to dynamical variables $h_\mu(t)$. The probability of any pattern $\{s_i\}$, given the current state of the latent variables, is

$$P(s_i | \{h_\mu(t)\}) = \frac{1}{Z(\{h_\mu(t)\})} \exp\left( -\eta \sum_{\mu=1}^{N_F} s_i J_{i\mu} h_\mu(t) - \epsilon s_i \right), \tag{2}$$

where the parameter $\eta$ controls the scaling of the variables and $\epsilon$ controls the overall activity level. We modeled each latent variable as an Ornstein-Uhlenbeck process with the time scale $\tau_F$ (see Materials and methods). Thus our model has four parameters: $\eta$ (input scaling), $\epsilon$ (activity threshold), $\tau_F$ (dynamical timescale), and $N_F$ (number of latent variables).

Distributions of avalanche size and avalanche duration within this model followed approximate power laws (*Figure 1C*; see Materials and methods). In the example shown ($N_F = 5$, $\tau_F = 10^4$, $\eta = 4$ and $\epsilon = 12$), we found exponents $\tau = 1.89 \pm 0.02$ (size) and $\alpha = 2.11 \pm 0.02$ (duration). Further, the average size of avalanches with fixed duration scaled as $S \sim D^\gamma$, with the fitted $\gamma_{\text{fit}} = 1.24 \pm 0.02$, in agreement with the predicted value $\gamma_{\text{pred}} = 1.24 \pm 0.02$. Thus, our model could generate avalanche scaling, at least for some parameter choices. In the following sections, we examine how avalanche scaling depends on model parameters ($N_F$, $\tau_F$, $\eta$ and $\epsilon$; see Table 2). We first focus on two sets of simulations: one set with $N_F = 1$ latent variable, which does not generate scaling under coarse-graining (*Morrell et al., 2021*), and one set with $N_F = 5$ latent variables, which can generate such scaling for some values of parameters $\tau_F$, $\eta$, and $\epsilon$ (*Morrell et al., 2021*; *Table 1*).

## Avalanche scaling depends on the number of latent variables

We analyzed avalanches from one- and five-variable simulations, each with fixed latent dynamical timescale ($\tau_F = 5 \times 10^3$ time steps; see *Table 2* for parameters). In the following sections, time is measured in simulation time steps, see Materials and methods for converting time steps to seconds. We used established methods for measuring empirical power laws (*Clauset et al., 2009*). The

**Table 2.** Simulation parameters for *Figure 2*.

| Parameter | Description | Value |
|---|---|---|
| $\epsilon$ | bias towards silence | $\epsilon = 8$ (for $N_F = 1$) or $\epsilon = 12$ (for $N_F = 5$) |
| $\eta$ | variance multiplier | $\eta = 4.0$ |
| $N_F$ | number of latent fields | $N_F = 1$ or $5$ |
| $\tau_F$ | latent field time constant | $\tau_F = 10^3, ... 10^5$ |
| $N$ | number of cells | $N = 1024$ |

minimum cutoffs for size ($S_{\min}$) and duration ($D_{\min}$) are indicated by vertical lines in *Figure 2*. For the population coupled to a single latent variable, the avalanche size distribution was not well fit by a power law (*Figure 2A*). With a sufficiently high minimum cut-off ($D_{\min}$), the duration distribution was approximately power-law (*Figure 2B*).

We next assessed whether the simulation produced crackling. If so, the value $\gamma_{\text{fit}}$ obtained by fitting $\bar{S}(D) \sim D^{\gamma_{\text{fit}}}$ would be similar to $\gamma_{\text{pred}} = \frac{\alpha-1}{\tau-1}$. In many cases, such as the one-variable example shown in *Figure 2C*, the full range of avalanche durations were not fit by a single power law. Therefore, we determined the largest range, over which a power law was a good fit to the simulated observations. In this case, slightly over two decades of apparent scaling were observed starting from avalanches with minimum duration slightly less than 100 time steps (*Figure 2C*), with a best-fit value of $\gamma_{fit} \in [1.69, 1.74]$. As we did not find a power-law in the size distribution, calculating $\gamma_{\text{pred}}$ is meaningless. If we do it anyway, we obtain $\gamma_{pred} = 0.83 \pm 0.03$ (yellow line in *Figure 2C*), which clearly deviates from the fitted value of $\gamma$. Thus, for the single latent dynamical variable model ($\tau_F = 5000$), power-law fits are poor, and there is no crackling.

The five-variable model produces a different picture. We now find avalanches, for which size and duration distributions are much better fit by power-law models starting from very low minimum cutoffs (*Figure 2D–E*, *Figure 2—figure supplement 2*). Average size scaled with duration, again over more than two decades, with $\gamma_{\text{fit}} = 1.27 \pm 0.03$, which was in close agreement with $\gamma_{\text{pred}} = 1.25 \pm 0.02$ (*Figure 2F*). Holding other parameters constant, we thus found that scaling relationships and crackling arise in the multi-variable model but not the single-variable model.

## Avalanche scaling depends on the time scale of latent variables

Based on simulations in the previous section, we surmised that the five-variable simulation generated scaling more readily due to creating an 'effective' latent variable that had slower dynamics than any individual latent variable. We reasoned that at any moment in time, the latent variable state $h_\mu(t)$ is a vector in the latent space. Because coupling to the latent variables is random throughout the population, only the length ($\sim \sqrt{N_F}$) and not the direction of this vector matters, and the timescale of changes in this length would be much slower than $\tau_F$, the timescale of each of the components $h_\mu(t)$. We therefore speculated that increasing the timescale of dynamics of the latent variables should eventually lead to scaling and crackling in the single-variable model as well as the five-variable one. To examine the dependence of avalanche scaling on this timescale, we simulated one-variable and five-variable networks at fixed $\eta$ and $\epsilon$ coupled to latent variables with the correlation time of their Ornstein-Uhlenbeck dynamics of $\tau_F \in [10^3, 10^5]$ time steps, spanning from a factor of 10 faster to a factor of 10 slower than the original $\tau_F$ in *Figure 1*. Simulations were replicated five times at each combination of parameters by drawing new latent variable coupling values ($J_{i\mu}$), as well as new latent variable dynamics and instances of neural firing. For simulations that passed the criteria to be fitted by power laws, we plot the fitted values of $\tau$, $\alpha$, $\gamma_{\text{fit}}$, and $\gamma_{\text{fit}} - \gamma_{\text{pred}}$ (*Figure 2G–J*). Missing points are those for which distributions did not pass the power law fit criteria.

In the single-variable model, best-fit exponents changed abruptly for latent variable timescale around $\tau_F = 10^4$ (*Figure 2G and H*), while in the five-variable model, exponents tended to increase gradually (*Figure 2G and H*, red). The discontinuity in the single-variable case reflected a change in the lower cutoff values in the power-law fits: size and duration distributions generated with faster latent dynamics could be fit reasonably well to a power law by using a high value of the lower cutoff (*Figure 2—figure supplement 3*). For time scales greater than $\sim 10^4$, the minimum cutoffs dropped, and the single-variable model generated power-law distributed avalanches and crackling (*Figure 2J*), similar to the five-variable model. In summary, in the latent dynamical variable model, introducing multiple variables generated scaling at faster timescales. However, by slowing the timescale of the latent dynamics, the model generated signatures of critical avalanche scaling for both multi- and single-variable simulations.

## Avalanche criticality, input scaling, and firing threshold

In the previous section, we found that a very slow single latent dynamical variable generated avalanche criticality in the simulation population. Thus, from now on, we simplify the model in order to characterize avalanche statistics across values of input scaling $\eta$ and firing threshold $\epsilon$. Specifically, we modeled a population of $N = 128$ neurons coupled to a single quasi-static latent variable. We

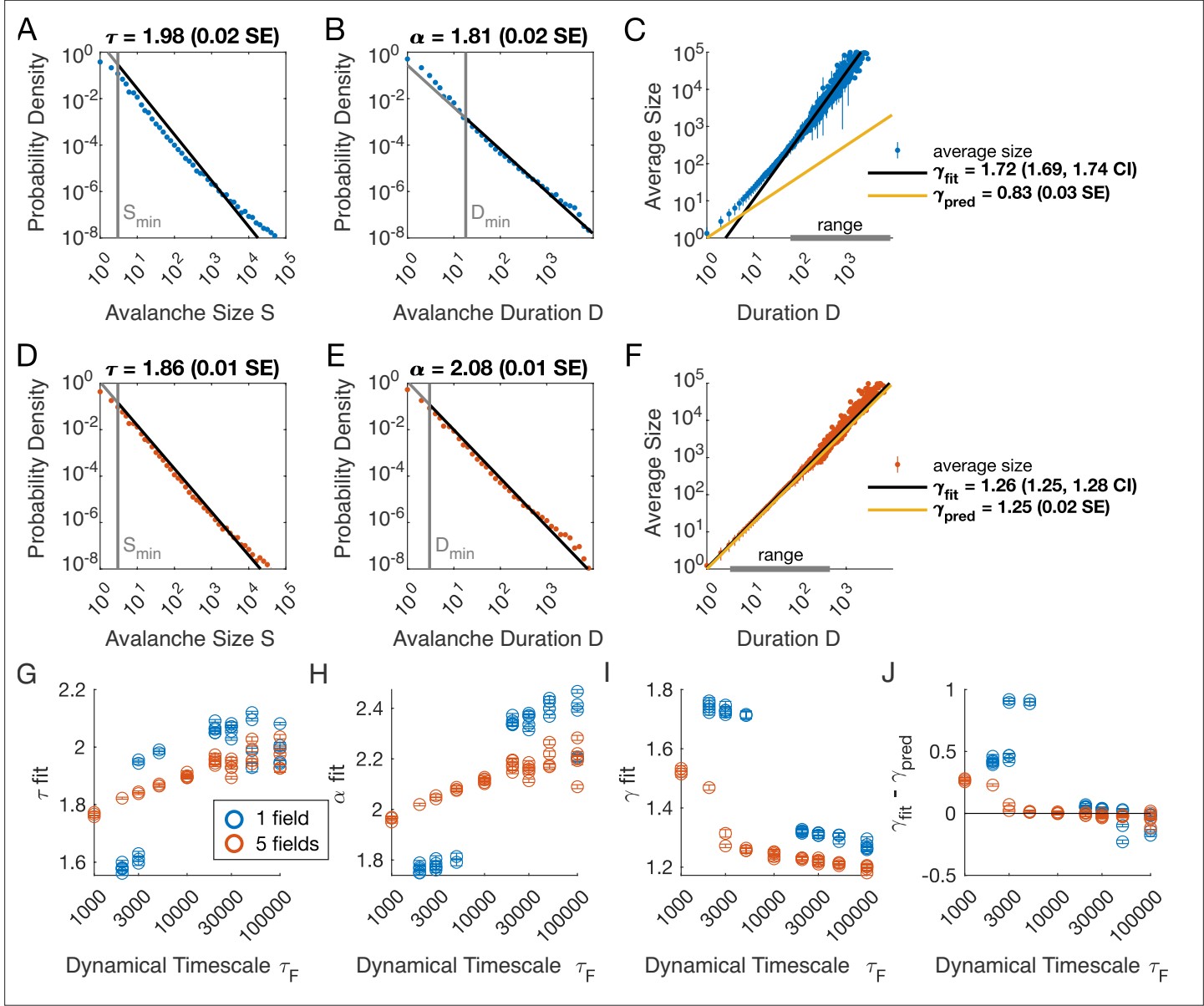

**Figure 2.** Multiple latent variables generate avalanche scaling at shorter timescales than a single latent variable. Simulated network is $N = 1024$ neurons. Other parameters used for simulations for this figure are found in *Table 2*. (**A-C**) Scaling analysis for one variable models where the dynamic timescale is equal to $5 \times 10^3$ time steps. (**A**) Distribution of avalanche sizes. MLE value of exponent for best-fit power law is $\tau = 1.98$ (0.02 SE), with lower cutoff indicated by the vertical line. (**B**) Distribution of avalanche duration. MLE value of $\alpha$ is 1.81 (0.02 SE). (**C**) Average size plotted against avalanche duration (blue points), with power-law fit (black line) and predicted relationship (yellow line) from MLE values for exponents in A and B. Gray bar on the horizontal axis indicates range, over which a power law with $\gamma = 1.72$ fits the data (see Materials and methods). (**D-F**) Analysis of avalanches from a simulation of a population coupled to five independent latent variables where the dynamic timescale is equal to $5 \times 10^3$ time steps. (**G**) Exponents $\tau$ for avalanche size distributions across timescales for one-variable (blue) and five-variable (red) simulations. Each circle is a simulation with independently drawn coupling parameters. Simulations had to show scaling over at least two decades to be included in panels (**G–J**). (**H**) Exponents $\alpha$ for avalanche duration distributions for simulations in G. (**I**) Fitted values of $\gamma$ for simulations in G. (**J**) Difference between fitted and predicted $\gamma$ values. Five-variable simulations produce crackling over a wider range of timescales than single-variable simulations.

The online version of this article includes the following figure supplement(s) for figure 2:

**Figure supplement 1.** Illustration of algorithm for determining $\tau$ and $\alpha$, using one variable example in *Figure 2*.

**Figure supplement 2.** Illustration of algorithm for determining $\tau$ and $\alpha$, using example in *Figure 2*, five latent variables.

**Figure supplement 3.** Illustration of algorithm for fitting the exponent $\gamma$ and determining the range, over which power law scaling of average size with duration is observed, using example in *Figure 2(A–D)*.

*Figure 2 continued on next page*

*Figure 2 continued*

**Figure supplement 4.** Illustration of algorithm for fitting the exponent $\gamma$ and determining the range over which power-law scaling of average size with duration is observed, using example in *Figure 2E–H*.

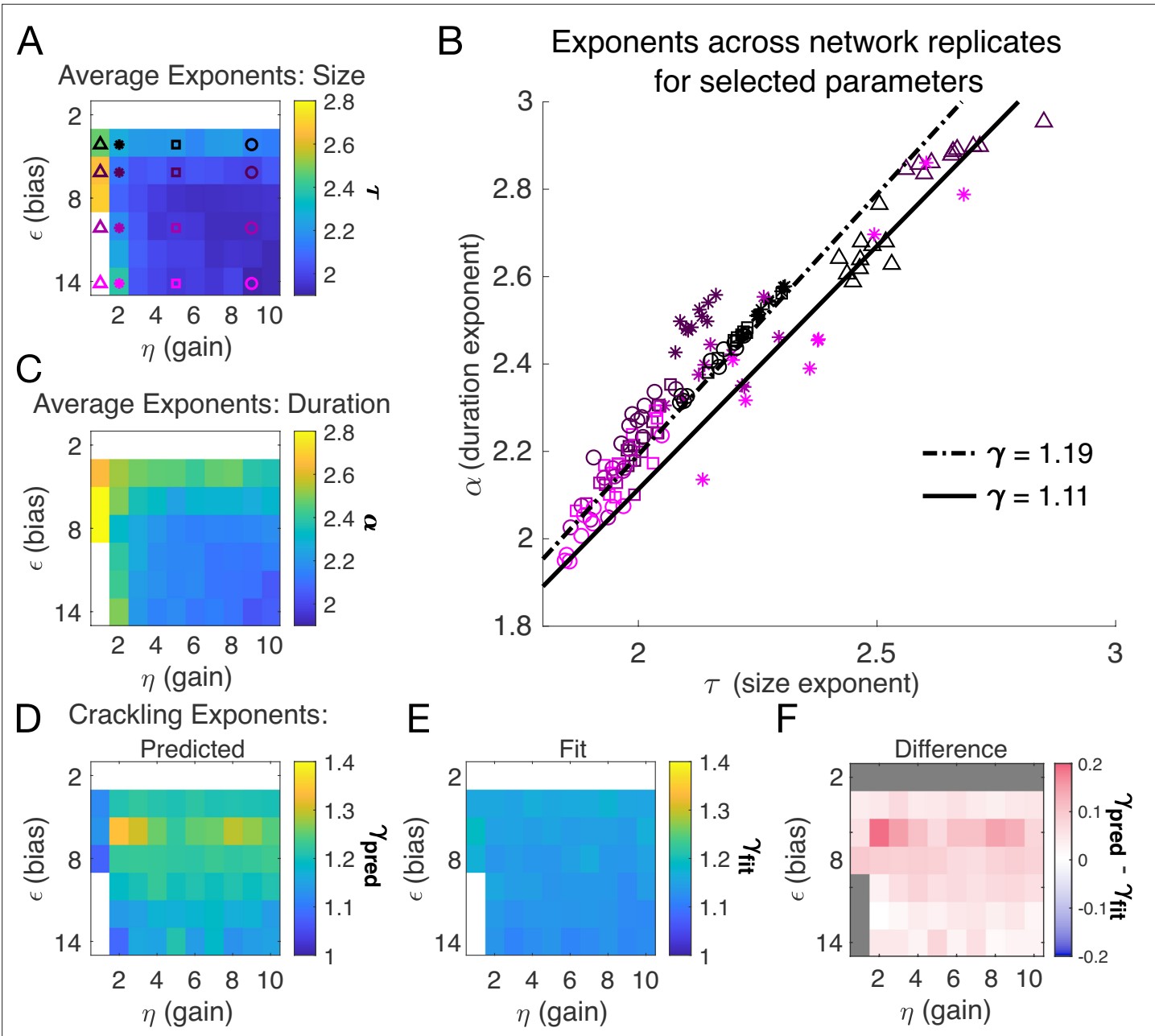

**Figure 3.** Exponents across network simulations for networks of $N = 128$ neurons. Each parameter combination $\eta, \epsilon$ was simulated for ten replicates, each time drawing randomly the couplings $J_i$, the latent variable values, and the neural activities. Other parameters in *Table 3*. (**A**) Average across replicates for the size exponent $\tau$. (**B**) Scatter plot of $\alpha$ vs. $\tau$ for each network replicate for parameter combinations indicated in A. Linear relationships between $\tau$ and $\alpha$, corresponding to the minimum and maximum values of $\gamma_{\text{fit}}$ from panel E, are shown to guide the eye. (**C**) Same as A, for duration exponent $\alpha$. (**D**) Predicted exponent, $\gamma_{\text{pred}}$, derived from A and C. (**E**) Value of $\gamma_{\text{fit}}$ from fit to $\bar{S} \sim D^\gamma$. (**F**) Difference between $\gamma_{\text{pred}}$ and $\gamma_{\text{fit}}$.

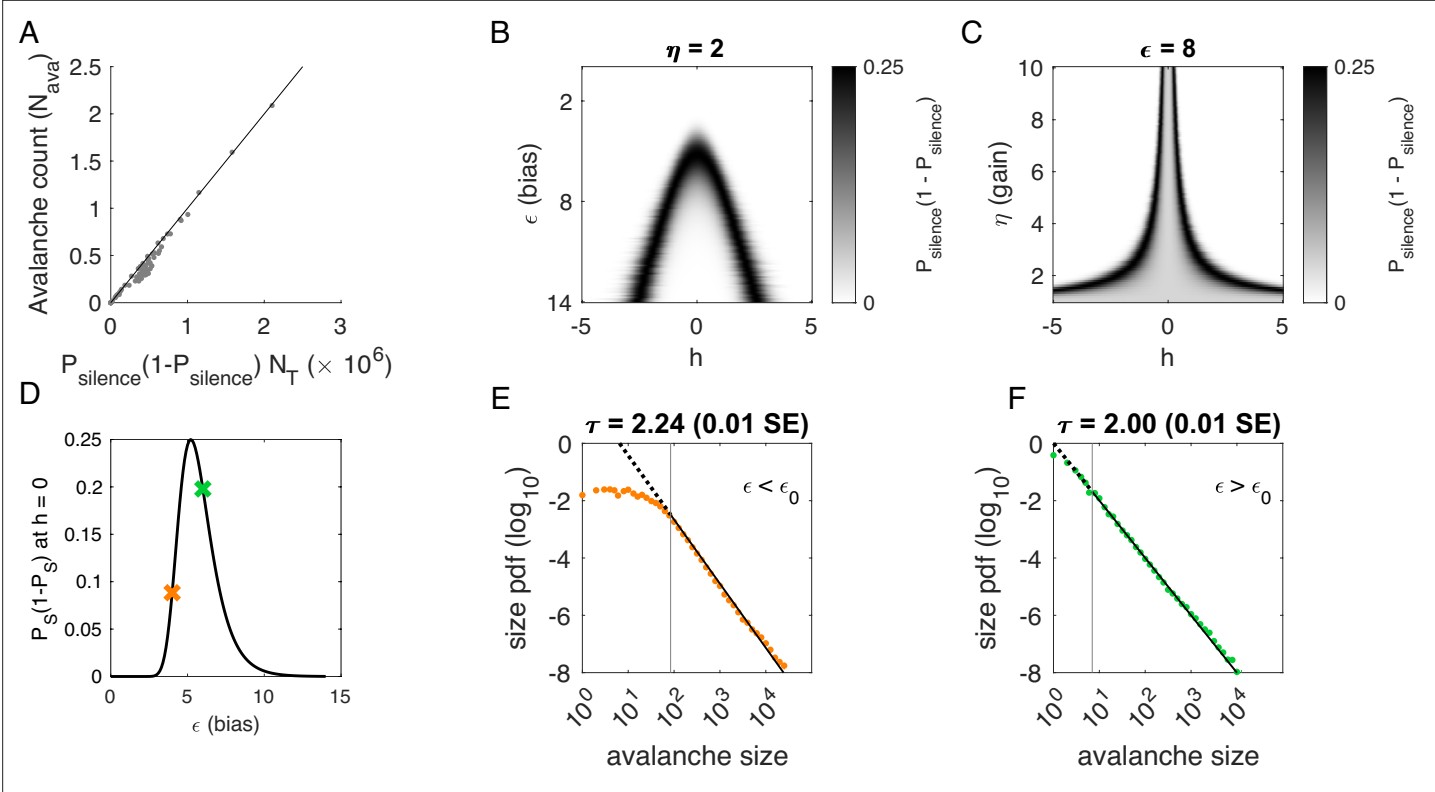

**Figure 4.** Avalanches in the latent dynamical variable model with a single quasistatic variable. Parameters in *Table 3*. (**A**) Number of avalanches in simulations from *Figure 3* as a function of the calculated probability of avalanches at fixed $\eta$ across values of $\epsilon$ and latent variable $h$. Line indicates equality. (**B**) Probability of avalanches with $\eta = 2$ across values of $\epsilon$ and $h$. The latent variable $h$ is normally distributed with mean 0 and variance 1. Where the distribution of $h$ overlaps with regions of high probability (black), avalanches occur. (**C**) Probability of avalanches at $\epsilon = 8$ across values of $\eta$ and $h$. Increasing $\eta$ narrows the range of $h$ that generates avalanches. (**D**) Probability of avalanches at $h = 0$ for a populations of 128 neurons (black line) and for a varying $\epsilon$. Size distributions corresponding to simulations marked by the green and orange crosses are in E, F. (**E**) Example of size distribution with $\epsilon < \epsilon_0$ (orange marker in D). Size cutoff is close to 100. (**F**) Example of size distribution with $\epsilon > \epsilon_0$ (green marker in D). Size cutoff is < 10.

The online version of this article includes the following figure supplement(s) for figure 4:

**Figure supplement 1.** Estimate of how long it takes to observe avalanche criticality at each combination of $\eta$ and $\epsilon$.

simulated $10^3$ segments of $10^4$ steps each and drew a new value of the latent variable ($h \sim N(0,1)$) for each segment. Ten replicates of the simulation were generated at each of the combinations of $\eta$ and $\epsilon$ (see Materials and methods).

Almost independent of $\eta$ and $\epsilon$, we found quality power law fits and crackling. *Figure 3* shows the average (across $n = 10$ network realizations) of the exponents extracted from size ($\tau$, *Figure 3A*) and duration ($\alpha$, *Figure 3C*) distributions. At small firing threshold ($\epsilon = 2$), we do not observe scaling because the system is always active, and all avalanches merge into one. At large firing threshold $\epsilon$ and low input scaling $\eta$, we do not observe scaling because activity is so sparse that all avalanches are small. At intermediate values of the parameters, the simulations generated plausible scaling relationships in size and duration. The difference between $\gamma_{\text{fit}}$ and $\gamma_{\text{pred}}$ was typically less than 0.1 (*Figure 4D–F*), which was consistent with previously reported differences between fit and predicted exponents (*Ma et al., 2019*). Thus, there appears to be no need for fine-tuning to generate apparent scaling in this model, at least in the limit of (near) infinite observation time. Wherever $\eta$ and $\epsilon$ generate avalanches, there are approximate power-law distributions and crackling.

To determine where avalanches occur, we derive the avalanche rate across values of the latent variable $h$. In the quasi-static model, the probability of an avalanche initiation is the probability of a transition from the quiet to an active state. Because all neurons are conditionally independent, this is $P_{\text{ava}} = P_{\text{silence}}(1 - P_{\text{silence}})$. Then the expected number of avalanches $\hat{N}_{\text{ava}}$ is obtained by integrating $P_{\text{ava}}$ over $h$ at each value of $\eta$ and $\epsilon$:

$$\hat{N}_{\text{ava}} = \int P_{\text{ava}}(\epsilon, \eta, h; J_i, N)p(h)dh = \int \prod_i \left(\frac{1}{1 + e^{-\eta J_i h - \epsilon}}\right)\left(1 - \prod_i \left(\frac{1}{1 + e^{-\eta J_i h - \epsilon}}\right)\right)p(h)dh, \quad (3)$$

where $p(h)$ is the standard normal distribution. This probability tracks the observed number of avalanches across simulations, **Figure 4A**.

To gain an intuition for the conditions under which avalanches occur, we show two slices of the avalanche probability, at fixed $\eta$ (**Figure 4B**) and at fixed $\epsilon$ (**Figure 4C**). Black regions indicate where avalanches are likely to occur. If, for a given value of $\epsilon$ and $\eta$, there is no overlap between high avalanche probability regions and the distribution of $h$, then there will be no avalanches. For large $\epsilon$, avalanches occur because neurons with large coupling to the latent variable ($\eta|J_i| >> 1$, recall $J_i \sim N(0, 1)$) are occasionally activated by a value of the latent variable $h$ that is sufficient to exceed $\epsilon$ (**Figure 4B**). Thus, the scaling parameter $\eta$ controls the value of $h$ for which avalanches occur most frequently (**Figure 4C**). As $\epsilon$ decreases, avalanches occur for smaller and smaller $h$ until avalanches primarily occur when $h = 0$.

To calculate the probability of avalanches, we must integrate over all values of $h$, but we can gain a qualitative understanding of which avalanche regime the system is in by examining the probability of avalanches at $h = 0$. At $h = 0$, the avalanche probability (see Materials and methods) is

$$P_{\text{ava}}(\epsilon, \eta, h = 0; J_i, N) = \left(\frac{1}{1 + e^{-\epsilon}}\right)^N \left(1 - \left(\frac{1}{1 + e^{-\epsilon}}\right)^N\right), \quad (4)$$

which is maximized at $\epsilon_0 = -\log(2^{1/N} - 1)$, independent of $J_i$ and $\eta$. After some algebra, we find that $\epsilon_0 \sim \log N$ for large $N$. The dependence on $N$ reflects that a larger threshold is required for larger networks: large networks ($N \to \infty$) are unlikely to achieve complete network silence, therefore preventing avalanches from occurring. Similarly, small networks ($N \sim 1$) are unlikely to fire consecutively and thus are unlikely to avalanche.

We plot $P_{\text{ava}}(\epsilon, \eta; J_i, N, h = 0)$ as a function of $\epsilon$ in **Figure 4D**. The peak at $\epsilon_0$ divides the space into two regions. For $\epsilon < \epsilon_0$, a power-law is only observed in the large-size avalanches, which are rare (**Figure 4E**, green). By contrast, when $\epsilon > \epsilon_0$, minimum size cutoffs are low (**Figure 4F**, orange). Both regions, $\epsilon < \epsilon_0$ and $\epsilon > \epsilon_0$, exhibit crackling noise scaling. If observation times are not sufficiently long (estimated in **Figure 4—figure supplement 1**), then scaling will not be observed in the $\epsilon < \epsilon_0$ region, whose scaling relations arise from rare events. Insufficient observation times may explain experiments and simulations where avalanche scaling was not found.

## Inferring the latent variable

Our analysis of $P_{\text{ava}}(\epsilon, \eta, h)$ at $h = 0$ suggested that there are two types of avalanche regimes: one with high activity and high minimum cutoffs in the power law fit (Type 1), and the other with lower activity and size cutoffs (Type 2). Further, when $P_{\text{ava}}$ drops to zero, avalanches disappear because the activity is too high or too low. We now examine how information about the value of the latent variables represented in the network activity relates to the activity type. To delineate these types, we calculated numerically $\epsilon^*(\eta)$, the value of $\epsilon$, for which the probability of avalanches is maximized, and the contours of $P_{\text{ava}}$ (**Figure 5A**). Curves for $\epsilon^*(\eta)$ and $\epsilon_0$ and $P_{\text{ava}} = 10^{-3}$ are shown in **Figure 5A and B**.

We expect that the more cells fire, the more information they would convey, until the firing rate saturates, and inferring the value of the latent variable becomes impossible. **Figure 5B** supports the prediction: generally, information is higher in regions with more activity (lower $\epsilon$, higher $\eta$), but only up to a limit: as $\epsilon \to 0$, information decreases. This decrease begins approximately where the probability of avalanches drops to nearly zero (dashed black lines, **Figure 5B–E**) because all of the activity merges into a few very large avalanches. In other words, the Type-1 avalanche region coincides with the highest information about the latent variable.

The critical brain hypothesis suggests that the brain operates in a critical state, and its functional role may be in optimizing information processing (**Beggs, 2008**; **Chialvo, 2010**). Under this hypothesis, we would expect the information conveyed by the network to be maximized in the regions we observe avalanche criticality. However, we see that critical regions do not always have optimal information transmission. In **Figure 5**, the region that displays crackling noise is that where avalanches exist ($P_{\text{ava}} > 0.001$), which corresponds to any $\eta$ value and $\epsilon \gtrsim 3$. This avalanche region encompasses

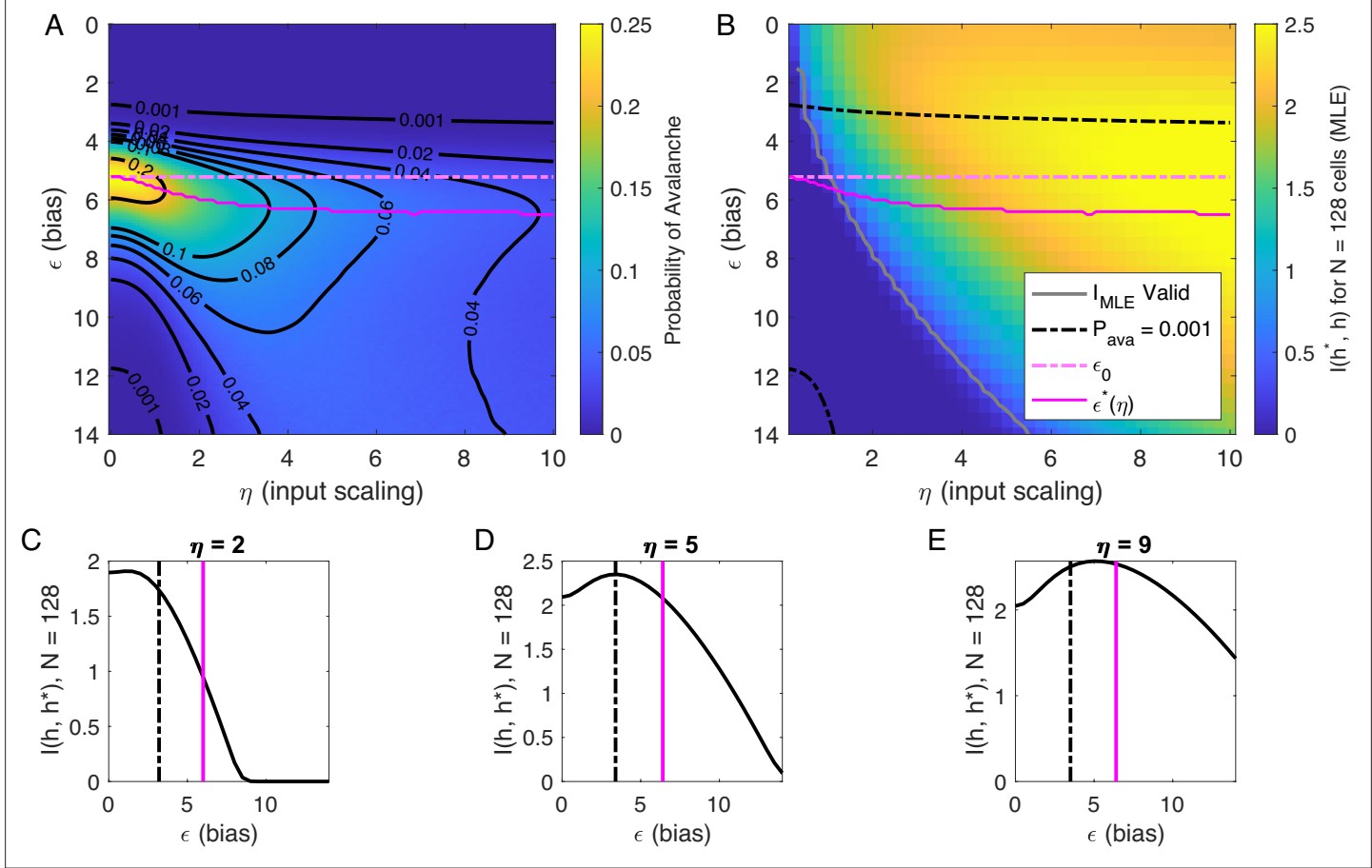

**Figure 5.** Information in the neural activity about the latent variable is higher in the low-$\epsilon$ avalanche region, compared to high-$\epsilon$ avalanche or high-rate avalanche-free activity. (**A**) Probability of avalanche per time step across values of $\eta$ and $\epsilon$. Solid magenta curve follows $\epsilon^*(\eta)$, the value of $\epsilon$ maximizing the probability of avalanches at fixed $\eta$. Dashed magenta line indicates $\epsilon_0$, calculated analytically, which matches $\epsilon^*$ at $\eta = 0$. (**B**) Information about latent variable, calculated from maximum likelihood estimate of $h$ using population activity. MLE approximation is invalid in the dark-blue region bounded by gray curve. Magenta line marks the maximum values of $P_{ava}$, reproduced from A. Dashed black curve indicates $P_{ava} = 0.001$. The highest information region falls between $\epsilon^*(\eta)$ and the contour for $P_{ava} = 0.001$. (**C - E**) Slices of B, showing $I_{MLE}(\epsilon)$ for $\eta = \{2, 5, 9\}$. Magenta and dashed black lines again indicate $\epsilon^*$ and $P_{ava} = 0.001$, respectively, as in B. Black dashed line marks the approximate boundary between the high-activity/no avalanche and the high-cutoff avalanche, and magenta line marks boundary between high-cutoff and low-cutoff avalanche regions.

both networks with high information transmission and networks with low information transmission. In summary, observing avalanche criticality in a system does not imply a high-information processing network state. However, the scaling can be seen at smaller cutoffs, and hence with shorter recordings, in the high-information state. This parallels the discussion by *Schwab et al., 2014*, who noticed that the Zipf's law always emerges in neural populations driven by quasi-stationary latent fields, but it emerges at smaller system sizes when the information about the latent variable is high.

## Discussion

Here, we studied systems with distributed, random coupling to latent dynamical variables and we found that avalanche criticality is nearly always observed, with no fine-tuning required. Avalanche criticality was surprisingly robust to changes in input gain and firing rate threshold. Loss of avalanche criticality could occur if the latent process was not well-sampled, either because the simulation was not long enough or the dynamics of the latent variables were too fast. Finally, while information about the latent variables in the network activity was higher where avalanches were generated compared

to when they were not, there was a range of information values across the critical avalanche regime. Thus, avalanche criticality alone was not a predictor of optimal information transmission.

## Explaining experimental exponents

A wide range of critical exponents has been found in ex vivo and in vivo recordings from various systems. For instance, the seminal work on avalanche statistics in cultured neuronal networks by *Beggs and Plenz, 2003* found size and duration exponents of 1.5 and 2.0 respectively, along with $\gamma = 2$, when time was discretized with a time bin equal to the average inter-event interval in the system. A subset of the in vivo recordings analyzed from anesthetized cat (*Hahn et al., 2010*) and macaque monkeys (*Petermann et al., 2009*) also exhibited a size distribution exponent close to 1.5. By contrast, a survey of many in vivo and ex vivo recordings found power-law size distributions with exponents ranging from 1 to 3 depending on the system (*Fontenele et al., 2019*). Separately, *Ma et al., 2019* reported recordings in freely moving rats with size exponents ranging from 1.5 to 2.7. In these recordings, when the crackling relationship held, the reported value of $\gamma$ was near 1.2 (*Fontenele et al., 2019*; *Ma et al., 2019*).

A model for generating avalanche criticality is a critical branching process (*Beggs and Plenz, 2003*), which predicts size and duration exponents of 1.5 and 2 and scaling exponent $\gamma$ of 2. However, there are alternatives: *Lombardi et al., 2023* showed that avalanche criticality may also be produced by an adaptive Ising model in the sub-critical regime, and in this case, the scaling exponent $\gamma$ was not 2 but close to 1.6. Our model, across the parameters we tested that produced exponents consistent with the scaling relationship, generated $\tau$ values that ranged from 1.9 to about 2.5. Across those simulations, we found values $\gamma$ within a narrow band from 1.1 to 1.3 (see *Figure 2I and J* and *Figure 3H*). While the exponent values our model produces are inconsistent with a critical branching process ($\gamma = 2$), they match the ranges of exponents estimated from experiments and reported by *Fontenele et al., 2019*. In this context, it might be useful to explore if our model and that of *Lombardi et al., 2023* might be related, with the adaptive feedback signal of the latter viewed as an effective, latent variable of the former.

A genuine challenge in comparing exponents estimated from different experiments with different recording modalities (spiking activity, calcium imaging, LFP, EEG, or MEG) arises from differences in spatial and temporal scale specific to a particular recording, as well as the myriad decisions made in avalanche analysis, such as defining thresholds or binning in time. Thus, one possible reason for differences in exponents across studies may derive from how the system is sub-sampled in space or coarse-grained in time, both of which systematically change exponents $\tau$ and $\alpha$ (*Beggs and Plenz, 2003*; *Shew et al., 2015*) and could account for differences in $\gamma$ (*Capek et al., 2023*). The model we presented here could be used as a test bed for examining how specific analysis choices affect exponents estimated from recordings.

A second possible explanation for differences in exponents is that different experiments study similar, but distinct biological phenomena. For instance, networks that were cultured in vitro may differ from those that were not, whether they are in vivo or ex vivo (i.e. brain slices), and sensory-processing networks may have different dynamics from networks with different processing demands. It is possible that certain networks develop connections between neurons such that they truly do produce dynamics that approximate a critical branching process, while other networks have different structure and resulting dynamics and thus can be better understood as coupled neurons receiving feedback (*Lombardi et al., 2023*) or as a system coupled to latent dynamical variables. This is especially true in sensory systems, where coupling to (latent) external stimuli in a way that the neural activity can be used to infer the stimuli is the reason for the networks' existence (*Schwab et al., 2014*).

## Relationship to past modeling work

Our model is not the first to produce approximate power-law size and duration distributions for avalanches from a latent variable process (*Touboul and Destexhe, 2017*; *Priesemann and Shriki, 2018*). In particular, *Priesemann and Shriki, 2018* derived the conditions, under which an inhomogeneous Poisson process could produce such approximate scaling. The basic idea is to generate a weighted sum of exponentially distributed event sizes, each of which are generated from a homogeneous Poisson process. How each process is weighted in this sum determines the approximate power-law exponent, allowing one to tune the system to obtain the critical values of 1.5 and 2.

Interestingly, this model did not generate non-trivial scaling of size with duration ($S \sim D^\gamma$). Instead, they found $\gamma = 1$, not the predicted $\gamma = 2$. Our results differ significantly, in that $\gamma$ was typically between 1.1 and 1.3 and it was nearly always close to the prediction from $\alpha$ and $\tau$. We speculate that this is due to nonlinearity in the mapping from latent variable to spiking activity, as doubling the latent field $h$ does not double the population activity, but doubling the rate of a homogeneous Poisson process does double the expected spike count. As biological networks are likely to have such nonlinearities in their responses to common inputs, this scenario may be more applicable to certain kinds of recordings.

## Summary

Latent variables – whether they are emergent from network dynamics (*Clark et al., 2023*; *Sederberg and Nemenman, 2020*) or derived from shared inputs – are ubiquitous in large-scale neural population recordings. This fact is reflected most directly in the relatively low-dimensional structure in large-scale population recordings (*Stringer et al., 2019*; *Pandarinath et al., 2018*; *Nieh et al., 2021*). We previously used a model based on this observation to examine signatures of neural criticality under a coarse-graining analysis and found that coarse-grained criticality is generated by systems driven by many latent variables (*Morrell et al., 2021*). Here, we showed that the same model also generates avalanche criticality, and that when information about the latent variables can be inferred from the network, avalanche criticality is also observed. Crucially, finding signatures of avalanche criticality required long observation times, such that the latent variable was well-sampled. Previous studies showed that Zipf's law appears generically in systems coupled to a latent variable that changes slowly relative to the sampling time, and that the Zipf's behavior is easier to observe in the higher information regime (*Schwab et al., 2014*; *Aitchison et al., 2016*). However, this also suggests that observation of either scaling at modest data set sizes indeed points to some fine-tuning — namely to the increase of the information in the individual neurons (and, since neurons in these models are conditionally independent, also in the entire network) about the value of the latent variables. In other words, one would expect a sensory part of the brain, if adapted to the statistics of the external stimuli, to exhibit all of these critical signatures at relatively modest data set sizes. In monocular deprivation experiments, when the activity in the visual cortex is transiently not adapted to its inputs, scaling disappears, at least for recordings of a typical duration, and is restored as the system adapts to the new stimulus (*Ma et al., 2019*). We speculate that the observed recovery of criticality by *Ma et al., 2019* could be driven by neurons adapting to the reduced stimuli state, for instance, by adjusting $\eta$ (input scaling) and $\epsilon$ (firing rate threshold).

Taken together, these results suggest that critical behavior in neural systems – whether based on the Zipf's law, avalanches, or coarse-graining analysis – is expected whenever neural recordings exhibit some latent structure in population dynamics and this latent structure can be inferred from observations of the population activity.

## Materials and methods
### Simulation of dynamic latent variable model

We study a model from *Morrell et al., 2021*, originally constructed as a model of large populations of neurons in mouse hippocampus. In the original version of the model, neurons are non-interacting, receiving inputs reflective of place-field selectivity as well as input current arising from a random projection from a small number of latent dynamical variables, representing inputs shared across the population of neurons that are not directly measured or controlled. In the current paper, we incorporate only the latent variables (no place variables), and we assume that every cell is coupled to every latent variable with some randomly drawn coupling strength.

The probability of observing a certain population state $\{s_i\}$ given latent variables $h_\mu(t)$ at time $t$ is

$$P(\{s_i\}|\{h_\mu\}) = \frac{1}{Z(\{h_\mu\})} e^{-H(\{s_i\},\{h_\mu\})},$$

(5)

where $Z$ is the normalization, and $H$ is the 'energy':

$$H = \sum_{i,\mu=1}^{N,N_f} \eta h_\mu(t) J_{i\mu} s_i + \epsilon s_i.$$

(6)

**Table 3.** Simulation parameters for *Figures 3 and 4*.

| Parameter | Description | Value |
|---|---|---|
| $\epsilon$ | bias towards silence | $\epsilon \in \{2, 4, ...14\}$ |
| $\eta$ | variance multiplier | $\eta \in \{1, 2, ...10\}$ |
| $N_F$ | number of latent fields | $N_F = 1$ |
| $\tau_F$ | latent field time constant | quasistatic |
| $N$ | number of cells | $N = 128$ |

The latent variables $h_\mu(t)$ are Ornstein-Uhlenbeck processes with zero mean, unit variance, and time constant $\tau_m$. Couplings $J_{i\mu}$ are drawn from the standard normal distribution.

Parameters for each figure are laid out in *Tables 1–3*. For the infinite time constant simulation, we draw a value $h \sim \mathcal{N}(0, 1)$ and simulate for 10000 time steps, then repeat for 1000 draws of $h$.

### Time step units

Most results were presented using arbitrary time units: all times (i.e. $\tau_F$ and avalanche duration $D$) are measured in units of an unspecified time step. Specifying a time bin converts the probability of firing into actual firing rates, in spikes per second, and this choice determines which part of the $\eta$-$\epsilon$ phase space is most relevant to a given experiment.

The time step is the temporal resolution at which activity is discretized, which varies from several to hundreds of milliseconds across different experimental studies (*Beggs and Plenz, 2003*; *Fontenele et al., 2019*; *Ma et al., 2019*). In physical units and assuming a bin size of 3 ms to 10 ms, our choice of $\eta$ and $\epsilon$ in *Figure 2* would yield physiologically realistic firing rate ranges (*Hengen et al., 2016*), with high-firing neurons reaching averages rates of 20-50 spikes/second and median firing-rate neurons around 1-2 spikes/second. The timescales of latent variables examined range from about 3 s to 3000 s, assuming 3-ms bins. Inputs with such timescales may arise from external sources, such as sensory stimuli, or from internal sources, such as changes in physiological state.

Simulations were carried out for the same number of time steps ($2\times10^6$), which would be approximately 1 to 2 'hours', a reasonable duration for in vivo neural recordings. Note that at large values of $\tau_F$, the latent variable space is not well sampled during this time period.

### Analysis of avalanche statistics

#### Setting the threshold for observing avalanches

In our model, we count avalanches as periods of continuous activity (in any subset of neurons) that is book-ended by time bins with no activity in the entire simulated neural network. For real neural populations of modest size, this method fails because there are no periods of quiescence. The typical solution is to set a threshold, and to only count avalanches when the population activity exceeds that threshold, with the hope that results are relatively robust to that choice. In our model, this operation is equivalent to changing $\epsilon$, which shifts the probability of firing up or down by a constant amount across all cells independent of inputs. Our results in *Figure 3* show that $\alpha$ and $\tau$ decrease as the threshold for detection is increased (equivalent to large $|\epsilon|$), but that the scaling relationship is maintained. The model predicts that $\gamma_{\text{pred}} - \gamma_{\text{fit}}$ would initially increase slightly with the detection threshold before decreasing back to near zero.

Following the algorithm laid out in *Clauset et al., 2009*, we fit power laws to the size and duration distributions from simulations generating avalanches. We use least-squares fitting to estimate $\gamma_{\text{fit}}$, the scaling exponent for size with duration, assessing the consistency of the fit across decades.

#### Reading power laws from data

We want, from each simulation, a quantification of the quality of scaling (how many decades, minimally) and an estimate of the scaling exponents ($\tau$ for the size distribution, $\alpha$ for the duration distribution). We first compile all avalanches observed in the simulation and for each avalanche, calculate its size (total activity across the population during the avalanche) and its duration (number of time bins).

Following the steps outlined by *Clauset et al., 2009*, we use the maximum-likelihood estimator to determine the scaling exponent. This is the solution to the transcendental equation

$$\frac{\zeta'(\hat{\alpha}, x_{\min})}{\zeta(\hat{\alpha}, x_{\min})} = -\frac{1}{n} \sum_{i=1}^{n} \ln x_i \qquad (7)$$

where $\zeta(\alpha, x_{\min})$ is the Hurwitz zeta function and $x_i$ are observations; that is, either the size or the duration of each avalanche $i$. For values of $x_{\min} < 6$, a numerical look-up table based on the built-in Hurwitz zeta function in the symbolic math toolbox was used (MATLAB2019b). For $x_{\min} > 6$ we use an approximation (*Clauset et al., 2009*),

$$\hat{\alpha} = 1 + n \left( \sum_i \ln \frac{x_i}{x_{\min} - \frac{1}{2}} \right)^{-1}. \qquad (8)$$

To determine $x_{\min}$, we computed the maximum absolute difference between the empirical cumulative density ($S(x)$) function and model's cumulative density function $P(x)$ (the Kolmogorov-Smirnov (KS) statistic; $D = \max_{x \geq x_{\min}} |S(x) - P(x)|$). The KS statistic was computed between for power-law models with scaling parameter $\hat{\alpha}$ and cutoffs $x_{\min}$. The value of $x_{\min}$ that minimizes the KS statistic was chosen. Occasionally the KS statistic had two local minima (as in *Figure 2—figure supplement 1*), indicating two different power-laws. In these cases, the minimum size and duration cutoffs were the smallest values that were within 10% of the absolute minimum of the KS statistic. Note that the statistic is computed for each model only on the power-law portion of the CDF (i.e. $x_i \geq x_{\min}$). We do not attempt to determine an upper cut-off value.

To assess the quality of the power-law fit, *Clauset et al., 2009* compared the empirical observations to surrogate data generated from a semi-parametric power-law model. The semi-parametric model sets the value of the CDF equal to the empirical CDF values up to $x = x_{\min}$ and then according to the power-law model for $x > x_{\min}$. If the KS statistic for the real data (relative to its fitted model) is within the distribution of the KS statistics for surrogate datasets relative to their respective fitted models, the power-law model was considered a reasonable fit.

Strict application of this methodology could give misleading results. Much of this is due to the loss of statistical power when the minimum cutoff is so high that the number of observations drops. For instance, in the simulations shown in *Figure 2*, the one-variable duration distribution passed the *Clauset et al., 2009* criterion, with a minimum KS statistic of 0.03 when the duration cutoff was 18 time steps. However, for the five-variable simulation in *Figure 2*, a power-law would be narrowly rejected for both size and duration, despite having much smaller KS statistics: for $\tau$, the KS statistic was 0.0087 (simulation range: 0.0008 to 0.0082; number of avalanches observed: 58,787) and for $\alpha$ it was 0.0084 (simulation range: 0.0011 to 0.0075). Below we discuss this problem in more detail.

## Determining range over which avalanche size scales with duration

For fitting $\gamma$, our aim was to find the longest sampled range, over which we have apparent power-law scaling of size with duration. Because our sampled duration values have linear spacing, error estimates are skewed if a naive goodness of fit criterion is used. We devised the following algorithm. First, the simulation must have at least one avalanche of size 500. We fit $S = cD^\gamma$ over one decade at a time. We chose as the lower duration cutoff the value of minimum duration, for which the largest number of subsequent (longer-duration) fits produced consistent fit parameters (*Figure 2—figure supplements 3 and 4*, top row). Next, with the minimum duration set, we gradually increased the maximum duration cut-off, and we determined whether there was a significant bias in the residual over the first decade of the fit. We selected the highest duration cutoff, for which there was no bias. Finally, over this range, we re-fit the power law relationship and extracted confidence intervals.

Our analysis focused on finding the apparent power-law relationship that held over the largest log-scale range. A common feature across simulation parameters ($\tau_F$, $N_F$) was the existence of two distinct power-law regimes. This is apparent in *Figure 2I*, which shows that when $N_F = 1$ at small $\tau_F$, the best-fit $\gamma$ (that showing the largest range with power-law-consistent scaling) is much larger ( 1.7), and then above $\tau_F \sim 3000$, the best-fit $\gamma$ drops to around 1.3.

## Statistical power of power-law tests

In several cases, we found examples of power-law fits that passed the rejection criteria commonly used to determine avalanche scaling relationships because of limited number of observations. A key example is that of the single latent variable simulation shown in *Figure 2B*, where we could not reject a power law for the duration distribution. Conversely, strict application of the surrogate criteria would reject a power law for distributions that were quantitatively much closer to a power-law (i.e. lower KS statistic), but for which we had many more observations and thus a much stronger surrogate test (*Figure 2*). This points to the difficulty of applying a single criterion to determining a power-law fit. In this work, we adhere to the criteria set forth in *Clauset et al., 2009*, with a modification to control for the unreasonably high statistical power of simulated data. Specifically, the number of avalanches used for fitting and for surrogate analysis was capped at 500,000, drawn randomly from the entire pool of avalanches.

Additionally, we found examples, in which a short simulation was rejected, but increasing the simulation time by a factor of five yielded excellent power-law fits. We speculate that this arises due to insufficient sampling of the latent space. These observations raise an important biological point. Simulations provide the luxury of assuming the network is unchanging for as long as the simulator cares to keep drawing samples. In a biological network, this is not the case. Over the course of hours, the effective latent degrees of freedom could change drastically (e.g. due to circadian effects [*Aton et al., 2009*], changes in behavioral state [*Fu et al., 2014*], plasticity [*Hooks and Chen, 2020*], etc.), and the network itself (synaptic scaling, firing thresholds, etc.) could be plastic (*Hengen et al., 2016*). All these factors can be modeled in our framework by determining appropriate cutoffs (in duration of recording, in time step sizes, for activity distributions) based on specific experimental timescales.

## Calculation of avalanche regimes

In the quasistatic model, we derive the dependence of the avalanche rate on $\eta$, $\epsilon$ and number of neurons $N$, finding that there are two distinct regimes, in which avalanches occur. Each time bin is independent, conditioned on the value of $h$. For an avalanche to occur, the probability of silence in the population (i.e. all $s_i = 0$) must not be too close to 0 (or there are no breaks in activity) or too close to 1 (or there is no activity). At fixed $h$, the probability of silence is

$$P_{\text{silence}}(\epsilon, \eta; J_i, N, h) = \prod_i \frac{1}{1 + \exp(-\eta J_i h + \epsilon)}. \tag{9}$$

An avalanche occurs when a silent time bin is followed by an active bin, which has probability $P_{\text{ava}}(\epsilon, \eta; J_i, N, h) = P_{\text{silence}}(1 - P_{\text{silence}})$.

## **Information calculation**

### Maximum-likelihood decoding

For large populations coupled to a single latent variable, we estimated the information between population spiking activity and the latent variable as the information between the maximum-likelihood estimator $h^*$ of the latent variable $h$ and the latent variable itself. This approximation fails at extremes of network activity levels (low or high).

Specifically, we approximated the log-likelihood of $h^*$ given $h_{\text{true}}$ near $h^*$ by $\log L(h - h^*) \approx \log L_{max} - \frac{1}{2} \frac{(h - h^*)^2}{\sigma_{h^*}^2}$. Thus we assume that $h^*$ is normally distributed about $h_{\text{true}}$ with variance $\sigma^2(h_{\text{true}})$. The variance is then derived from the curvature of the log-likelihood at the maximum. The information between two Gaussian variables, here $P(h^*|h) = N(h, \sigma_{h^*}^2)$ and $p(h) = N(0, 1)$, is

$$I(h; \bar{s}_{i,T}) \approx \frac{1}{2} \left\langle \log \frac{T}{\sigma_{h_{\text{true}}}^2} \right\rangle_{h_{\text{true}}}, \tag{10}$$

where the average is taken over $h_{\text{true}} \sim N(0, 1)$.

Given a set of $T$ observations of the neurons $\{s_i\}$, the likelihood is

$$P(\{s_i\}_t|h) = \prod_{i,t}^{N,T} P(s_i|h) = \prod_{i,t}^{N,T} \frac{e^{-\eta s_i J_i h - \epsilon s_i}}{1 + e^{-(\eta J_i h + \epsilon)}}. \tag{11}$$

Maximizing the log likelihood gives the following condition:

$$0 = \frac{\partial(\log P)}{\partial h}\Big|_{h^*} = \frac{\partial}{\partial h}\left(\sum_{i,t}\left((-\eta s_i J_i h - \epsilon s_i) - \log(1 + e^{-(\eta J_i h + \epsilon)})\right)\right)\Big|_{h^*} \tag{12}$$

$$= \sum_i -\eta \bar{s}_i J_i T + \frac{T J_i \eta}{1 + e^{\eta J_i h^* + \epsilon}}, \tag{13}$$

where $\bar{s}_i = \frac{1}{T}\sum_t s_{it}$ is the average over observations $t$. The uncertainty in $h^*$ is $\sigma_h$, which was calculated from the second derivative of the log likelihood:

$$\frac{1}{\sigma_{h^*}^2} = -\frac{\partial^2(\log P)}{\partial h^2} \tag{14}$$

$$= -\frac{\partial}{\partial h}\left(\sum_i -\eta \bar{s}_i J_i T + \frac{T J_i \eta}{1 + e^{\eta J_i h + \epsilon}}\right)\Big|_{h^*} \tag{15}$$

$$= \sum_i \frac{T(\eta J_i)^2 e^{\eta J_i h^* + \epsilon}}{(1 + e^{\eta J_i h^* + \epsilon})^2} \tag{16}$$

$$= \sum_i \frac{T(\eta J_i)^2}{4\cosh^2(\frac{\eta J_i h^* + \epsilon}{2})}. \tag{17}$$

This expression depends on the observations $\bar{s}_i$ only through the maximum-likelihood estimate $h^*$. When $h^* \to h_{\text{true}}$, then the variance is

$$\frac{1}{\sigma_{h^*}^2} = \sum_i \frac{T(\eta J_i)^2}{4\cosh^2(\frac{\eta J_i h_{\text{true}} + \epsilon}{2})} \equiv \frac{T}{\sigma_{h_{\text{true}}}^2}. \tag{18}$$

To generate *Figure 5*, we evaluated *Equation 10* using *Equation 18*.

## Code availability

Simulation code was adapted from our previous work (*Morrell et al., 2021*). Code to run simulations and perform analyses presented in this paper is uploaded as *Source code 1* and also available from https://github.com/ajsederberg/avalanche (copy archived at *Sederberg, 2024*).

## Acknowledgements

IN was supported in part by the Simons Foundation Investigator program, the Simons-Emory Consortium on Motor Control, NSF grant BCS/1822677 and NIH grant 2R01NS084844. AS was supported in part by NIH grant 1RF1MH130413-01 and by startup funds from the University of Minnesota Medical School. The authors acknowledge the Minnesota Supercomputing Institute (MSI) at the University of Minnesota for providing resources that contributed to the research results reported within this paper.

## Additional information

### Competing interests
Audrey Sederberg: Reviewing editor, eLife. The other authors declare that no competing interests exist.

## Funding

| Funder | Grant reference number | Author |
| --- | --- | --- |
| Simons Foundation | Simons-Emory Consortium on Motor Control | Ilya Nemenman |
| National Science Foundation | BCS/1822677 | Ilya Nemenman |
| National Institute of Neurological Disorders and Stroke | 2R01NS084844 | Ilya Nemenman |
| National Institute of Mental Health | 1RF1MH130413 | Audrey Sederberg |
| Simons Foundation | Investigator Program | Ilya Nemenman |

The funders had no role in study design, data collection and interpretation, or the decision to submit the work for publication.

## Author contributions

Mia C Morrell, Audrey Sederberg, Conceptualization, Software, Formal analysis, Validation, Investigation, Visualization, Methodology, Writing – original draft, Writing – review and editing; Ilya Nemenman, Conceptualization, Software, Formal analysis, Validation, Investigation, Visualization, Methodology, Writing – review and editing

## Author ORCIDs

Audrey Sederberg ⓘ https://orcid.org/0000-0003-4458-3773

Joint Public Review https://doi.org/10.7554/eLife.89337.3.sa1
Author Response: https://doi.org/10.7554/eLife.89337.3.sa2

---

# Additional files

## Supplementary files
- MDAR checklist
- Source code 1. Code (Python and MATLAB) used to run simulations, analyses, and calculations.

## Data availability

The current manuscript is a computational study, so no data were generated for this manuscript. Modelling code is uploaded as *Source code 1*.

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
