## [Editor Report · eLife assessment]

This paper provides a simple example of a neural-like system that displays criticality, but not for any deep reason; it's just because a population of neurons are driven (independently!) by a slowly varying latent variable, something that is common in the brain. Moreover, criticality does not imply optimal information transmission (one of its proposed functions). The work is likely to have an **important** impact on the study of criticality in neural systems and is **convincingly** supported by the experiments presented.

---

## [Referee Report · Joint Public Review]

This paper shows that networks of binary neurons can exhibit power law behavior (including "crackling", which refers to a particular relationship among the power law exponents) without fine tuning. If, as is standard, we equate power law behavior to criticality, then criticality can arise in networks of neurons without fine tuning. The network model used to show this was extremely simple: a population of completely uncoupled neurons was driven by a small number of slowly varying "hidden" variables (either 1 or 5). This caused the firing rate of every neuron to change slowly over time, in a correlated fashion. Criticality was observed over a large range of couplings, time constants, and average firing rates.

This paper is extremely important in light of the hypothesis that criticality in the brain is both special, in the sense that it requires fine tuning, and that it leads to optimal information processing. As mentioned above, this paper shows that fine tuning is not required. It also shows that criticality does not imply optimal information transmission. This does not, of course, rule out the above "critical brain" hypothesis. But it does show that simply observing power law behavior is not enough to draw conclusions about either fine tuning or function.

These authors are not the first to show that slowly varying firing rates can give rise to power law behavior (see, for example, Touboul and Destexhe, 2017; Priesemann and Shriki, 2018). However, to our knowledge they are the first to show crackling, and to compute information transmission in, and out of, the critical state.

**References:**

Touboul and Destexhe, 2017: Touboul J, Destexhe A. Power-law statistics and universal scaling in the absence of criticality. Phys Rev E. 2017 95:012413, 2017.

Priesemann and Shriki, 2018: Priesemann V, Shriki O. PLOS Comp. Bio. 14:1-29, 2018.

---

## [Author Response]

The following is the authors’ response to the original reviews.

**Joint Public Review:**
[…] While this does not rule out criticality in the brain, it decidedly weakens the evidence for it, which was based on the following logic: critical systems give rise to power law behavior; power law behavior is observed in cortical networks; therefore, cortical networks operate near a critical point. Given, as shown in this paper, that power laws can arise from noncritical processes, the logic breaks. Moreover, the authors show that criticality does not imply optimal information transmission (one of its proposed functions). This highlights the necessity for more rigorous analyses to affirm criticality in the brain. In particular, it suggests that attention should be focused on the question "does the brain implement a dynamical latent variable model?".These authors are not the first to show that slowly varying firing rates can give rise to power law behavior (see, for example, Touboul and Destexhe, 2017; Priesemann and Shriki, 2018). However, to our knowledge they are the first to show crackling, and to compute information transmission in the critical state.

We thank the reviewers for their thoughtful assessment of our paper.

We would push back on the assessment that our model ‘has nothing to do with criticality,’ and that we observed ‘signatures of criticality [that] emerge through fundamentally non-critical mechanisms.’ This assessment partially stems from the definition of criticality provided in the Public Comment, that ‘criticality is a very specific set of phenomena in physics in which fundamentally local interactions produce unexpected long-range behavior.’

Our disagreement is largely focused on this definition, which we do not think is a standard definition. Taking the favorite textbook example, the Ising model, criticality is characterized by a set of power-law divergences in thermodynamic quantities (e.g., susceptibility, specific heat, magnetization) at the critical temperature, with exponents of these power laws governed by scaling laws. It is not defined by local interactions. All-to-all Ising model is generally viewed as showing a critical behavior at a certain temperature, even though interactions there are manifestly non-local. It is possible that, by “local” in the definition, the Public Comment meant that interactions are “collective” and among microscopic degrees of freedom. However, that same all-to-all Ising model is mathematically equivalent to the mean-field model, where criticality is achieved through large fluctuations of the mean field, but not through microscopic interactions.

More commonly, criticality is defined by power laws and scaling relationships that emerge at a critical value of a parameter(s) of the system. That is, criticality is defined by its signatures. What is crucial in all such definitions is that this atypical, critical state requires fine tuning. For example, in the textbook example of the Ising model, a parameter (the temperature) must be tuned to a critical value for critical behavior to appear. In the branching process model that generates avalanche criticality, criticality requires tuning m=1. The key result of our paper is that all signatures expected for avalanche criticality (power laws, crackling, and, as shown below, estimates of the branching rate m), and hence the criticality itself, appear without fine-tuning.

As we discussed in our introduction, there are a few other instances of signatures of criticality (and hence of criticality itself) emerging without fine-tuning. The first we are aware of was the demonstration of Zipf’s Law (by Schwab, et al. 2014, and Aitchison et al. 2016), a power-law relationship between rank and frequency of states, which was shown to emerge generically in systems driven by a broadly distributed latent variable. A second example, arising from applications of coarse-graining analysis to neural data (cf., Meshulam et al. 2019; also, Morales et al., 2023), was demonstrated in our earlier paper (Morrell et al. 2021). Thus, here we have a third example: the model in this paper generates signatures of criticality in the statistics of avalanches of activity, and it does so without fine-tuning (cf., Fig. 2-3).

The rate at which these ‘criticality without fine-tuning' examples are piling up may inspire revisiting the requirement of fine-tuning in the definition of criticality, and our ongoing work (Ngampruetikorn et al. 2023) suggests that criticality may be more accurately defined through large fluctuations (variance > 1/N) rather than through fine-tuning or scaling relations.

**References:**

Schwab DJ, Nemenman I, Mehta P. “Zipf’s Law and Criticality in Multivariate Data without FineTuning.” Phys Rev Lett. 2014 Aug; doi::101103/PhysRevLett.113.068102,Aitchison L, Corradi N, Latham PE. “Zipf’s Law Arising Naturally When There Are Underlying, Unobserved Variables.” PLOS Computational biology. 2016 12; 12(12):1-32. doi:10.1371/journal.pcbi.1005110Meshulam L, Gauthier JL, Brody CD, Tank DW, Bialek W. “Coarse Graining, Fixed Points, and Scaling in a Large Population of Neurons.” Phys Rev Lett. 2019 Oct; doi: 10.1103/PhysRevLett.123.178103.Morales GB, di Santo S, Muñoz MA. “Quasiuniversal scaling in mouse-brain neuronal activity stems from edge-of-instability critical dynamics.” Proceedings of the National Academy of Sciences. 2023; 120(9):e2208998120.Morrell MC, Sederberg AJ, Nemenman I. “Latent Dynamical Variables Produce Signatures of Spatiotemporal Criticality in Large Biological Systems.” Phys Rev Lett. 2021 Mar; doi: 10.1103/PhysRevLett.126.118302.Ngampruetikorn, V., Nemenman, I., Schwab, D., “Extrinsic vs Intrinsic Criticality in Systems with Many Components.” arXiv: arXiv:2309.13898 [physics.bio-ph]

Major comments:1. For many readers, the essential messages of the paper may not be immediately clear. For example, is the paper criticizing the criticality hypothesis of cortical networks, or does the criticism extend deeper, to the theoretical predictions of "crackling" relationships in physical systems as they can emerge without criticality? Statements like "We show that a system coupled to one or many dynamical latent variables can generate avalanche criticality ..." could be misinterpreted as affirming criticality. A more accurate language is needed; for instance, the paper could state that the model generates relationships observed in critical systems. The paper should provide a clearer conclusion and interpretation of the findings in the context of the criticality hypothesis of cortical dynamics.

Please see the response to the Public Review, above. To clarify the essential message that the dynamical latent variable model produces avalanche criticality without fine-tuning, we have made revisions to the abstract and introduction. This point was already made in the discussion (first sentence).

Key sentences changed in the abstract:

"… We find that populations coupled to multiple latent variables produce critical behavior across a broader parameter range than those coupled to a single, quasi-static latent variable, but in both cases, avalanche criticality is observed without fine-tuning of model parameters. … Our results suggest that avalanche criticality arises in neural systems in which activity is effectively modeled as a population driven by a few dynamical variables and these variables can be inferred from the population activity."

In the introduction, we changed the final sentence to read:

"These results demonstrate how criticality in neural recordings can arise from latent dynamics in neural activity, without need for fine-tuning of network parameters."

2. On lines 97-99, the authors state that "We are agnostic as to the origin of these inputs: they may be externally driven from other brain areas, or they may arise from recurrent dynamics locally". This idea is also repeated at the beginning of the Summary section. Perhaps being agnostic isn't such a good idea: it's possible that the recurrent dynamics is in a critical regime, which would just push the problem upstream. Presumably you're thinking of recurrent dynamics with slow timescales that's not critical? Or are you happy if it's in the critical regime? This should be clarified.

We have amended this sentence to clarify that any latent dynamics with large fluctuations would suffice:

”We are agnostic as to the origin of these inputs: they may be externally driven from other brain areas, or they may arise from large fluctuations in local recurrent dynamics.”

3. Even though the model in Equation 2 has been described in a previous publication and the Methods section, more details regarding the origin and justification of this model in the context of cortical networks would be helpful in the Results section. Was it chosen just for simplicity, or was there a deeper reason?

This model was chosen for its simplicity: there are no direct interactions between neurons, coupling between neurons and latent variables is random, and simulation is straightforward. More complex latent dynamics or non-random structure in the coupling matrices could have been used, but our aim was to explore this model in the simplest setting possible.

We have revised the Results (“Avalanche scaling in a dynamical latent variable model,” first paragraph) to justify the choice of the model:

"We study a model of a population of neurons that are not coupled to each other directly but are driven by a small number of dynamical latent variables -- that is, slowly changing inputs that are not themselves measured (Fig.A). We are agnostic as to the origin of these inputs: they may be externally driven from other brain areas, or they may arise from large fluctuations in local recurrent dynamics. The model was chosen for its simplicity, and because we have previously shown that this model with at least about five latent variables can produce power laws under the coarse-graining analysis Morrell2021."

We have added the following to the beginning of the Methods section expanding on the reasons for this choice:

"We study a model from Morrell 2021, originally constructed as a model of large populations of neurons in mouse hippocampus. Neurons are non-interacting, receiving inputs reflective of place-field selectivity as well as input current arising from a random projection from a small number of dynamical latent variables, representing inputs shared across the population of neurons that are not directly measured or controlled. In the current paper, we incorporate only the latent variables (no place variables), and we assume that every cell is coupled to every latent variable with some randomly drawn coupling strength."

4. The Methods section (paragraph starting on line 340) connects the time scale to actual time scales in neuronal systems, stating that "The timescales of latent variables examined range from about 3 seconds to 3000 seconds, assuming 3-ms bins". While bins of 3 ms are relevant for electrophysiological data from LFPs or high-density EEG/MEG, time scales above 10 seconds are difficult to generate through biophysically clear processes like ionic channels and synaptic transmission. The paper suggests that slow time scales of the latent variables are crucial for obtaining power law behavior resembling criticality. Yet, one way to generate such slow time scales is via critical slowing down, implying that some brain areas providing input to the network under study may operate near criticality. This pushes the problem toward explaining the criticality of those external networks. Hence, discussing potential sources for slow time scales in latent variables is crucial. One possibility you might want to consider is sources external to the organism, which could easily have time scales in the 1-24 hour range.

As the reviewers note, it is a possibility that slow timescales arise from some other brain area in which dynamics are slow due to critical dynamics, but many other plausible sources exist. These include slowly varying sensory stimuli or external sources, as suggested by the reviewers. It is also possible to generate “effective” slow dynamics from non-critical internal sources. One example, from recordings in awake mice, is the slow change in the level of arousal that occurs on the scale of many seconds to minutes. These changes arise from release of neuromodulators that have broad effects on neural populations and correlations in activity (for a focused review, see Poulet and Crochet, 2019).

We have added the following sentence to the Methods section where timescales of latent variables was discussed:

"The timescales of latent variables examined range from about $3$ seconds to $3000$ seconds, assuming $3$-ms bins. Inputs with such timescales may arise from external sources, such as sensory stimuli, or from internal sources, such as changes in physiological state."

5. It is common in neuronal avalanche analysis to calculate the branching parameter using the ratio of events in consecutive bins. Near-critical systems should display values close to 1, especially in simulations without subsampling. Including the estimated values of the branching parameter for the different cases investigated in this study could provide more comprehensive data. While the paper acknowledges that the obtained exponents in the model differ from those in a critical branching process, it would still be beneficial to offer the branching parameter of the observed avalanches for comparison.

The reviewers requested that the branching parameter be computed in our model. We point out that, for the quasi-stationary latent variables (as in Fig. 3), a branching parameter of 1 is expected because the summed activity at time t+k is, on average, equal to the summed activity at time t, regardless of k. Numerics are consistent with this expectation. Following the methodology for an unbiased estimate of the branching parameter from Wilting and Priesemann (2018), we checked an example set of parameters (epsilon = 8, eta = 3) for quasi-stationary latent fields. We found that the naïve (biased) estimate of the branching parameter was 0.94, and that the unbiased estimator was exp(−1.4⋅10−8) ≈ 0.999999986.

For faster time scales, it is no longer true that summed activity is constant over time, as the temporal correlations in activity decay exponentially. Using the five-field simulation from Figure 2, we calculated the branching parameter for several values of tau. The biased estimates of m are 0.76 (τ=50), 0.79 (τ=500), and 0.79 (τ=5000). The corrected estimates are 0.98 (τ=50), 0.998 (τ=500), and 0.9998 (τ=5000).

6. In the Discussion (l 269), the paper suggests potential differences between networks cultured in vitro and in vivo. While significant differences indeed exist, it's worth noting that exponents consistent with a critical branching process have also been observed in vivo (Petermann et al 2009; Hahn et al. 2010), as well as in large-scale human data.

We thank the reviewers for pointing out these studies, and we have added the missing one (Hahn et al. 2010) to our reference list. The following was added to the discussion, in the section “Explaining Experimental Exponents:”

"A subset of the in vivo recordings analyzed from anesthetized cat (Hahn et al. 2010) and macaque monkeys (Petermann et al. 2009) exhibited a size distribution exponent close to 1.5."

Along these lines, we noted two additional studies of high relevance that have been published since our initial submission (Capek et al. 2023, Lombardi et al. 2023), and we have added these references to the discussion of experimental exponents.

Minor comments:1. The term 'latent variable' should be rigorously explained, as it is likely to be unfamiliar to some readers.

Sentences and clauses have been added to the Introduction, Results and the Methods to clarify the term:

Intro: “Numerous studies have reported relatively low-dimensional structure in the activity of large populations of neurons [refs], which can be modeled by a population of neurons that are broadly and heterogeneously coupled to multiple dynamical latent (i.e., unobserved) variables.”

Results: “We studied a population of neurons that are not coupled to each other directly but are driven by a small number of dynamical latent variables -- that is, slowly changing inputs that are not themselves measured.”

Methods: “Neurons are non-interacting, receiving inputs reflective of place-field selectivity as well as input current reflecting a random projection from a small number of dynamical latent variables, representing inputs shared across the population of neurons that are not directly measured.”

2. There's a relatively important typo in the equations: Eq. 2 and Eq. 6 differ by a minus sign in the exponent. Eqs. 3 and 4 use the plus sign, but epsilon_0 on line 198 uses the minus sign. All very confusing until we figured out what was going on. But easy to fix.

Thank you for catching this. We have made the following corrections:

1. Figures adopted the sign convention that epsilon > 0, with larger values of epsilon decreasing the activity level. Signs in Eqs. 3 and 4 have been corrected to match.

2. Equation 5 was missing a minus sign in front of the Hamiltonian. Restoring this minus sign fixed the discrepancy between 2 and 6.

3. In Eq. 7, the left hand side is zeta'/zeta', which is equal to 1. Maybe it should be zeta'/zeta? Fixed, thank you.

Additional comments:The authors are free to ignore these; they are meant to improve the paper.

We are extremely grateful for the close reading of our paper and note the actions taken below.

1. We personally would not use the abbreviation DLV; we find abbreviations extremely hard to remember. And DLV is not used that often.

Done, thank you for the suggestion.

2. l 198: epsilon_0 = -log(2^{1/N}-1) was kind of hard to picture -- we had to do a little algebra to make sense of it. Why not write e^{-epsilon_0} = 2^{1/N}-1 \approx log(2)/N, which in turn implies that epsilon_0 ~ log(N)?

Thank you, good point. We have added a sentence now to better explain:

"...which is maximized at $\epsilon_0 = - \log (2^{1/N} - 1)$, independent of $J_i$ and $\eta$. After some algebra, we find that $\epsilon_0 \sim \log N$ for large $N$."

3. Typo on l 202: "We plot P_ava as a function of epsilon in Fig. 4B". 4B  4D.

Done

4. It would be easier on the reader if the tables were all in one place. It would be even nicer to put the parameters in the figure captions. Or at least N; that one is kind of important.

Table placement was a Latex issue, which we have now fixed. We also have included links between tables and relevant figures and indicated network size.

5. What's x_i in Eqs. 7 and 8?

We added a sentence of explanation. These are the individual observations of avalanche sizes or durations, depending on what is being fit.

6. The latent variables evolve according to an Ornstein-Uhlenbeck process. But we might equally expect oscillations or non-normal behavior coupling dynamical modes, and these are likely to give different behavior with respect to avalanches. It might be worth commenting on this.7. The model assumes a normal distribution of the coupling strengths between the latent variables and the binary units. Discussing the potential effects of different types of random coupling could provide interesting insights.

Both 6 and 7 are interesting questions. At this point, we could speculate that the main results would be qualitatively unchanged, provided dynamics are sufficiently slow and that the distribution of coupling strengths is sufficiently broad (that is, there is variance in the coupling matrix across individual neurons). Further studies would be needed to make these statements more precise.

8. In Fig 1, tau_f = 1E4 whereas in Fig 2 tau_f = 5E3. Why the difference?

For Figure 1, we chose a set of parameters that gave clear scaling. In Figure 2, we saw some value in showing more than one example of scaling, hence different parameters for the examples in Fig 2 than Fig 1. Note that the Fig 1 simulations are represented in Fig. 2 G-J, as the 5-field simulation with tau_F = 1e4.